# A Pictorial Essay Describing the CT Imaging Features of COVID-19 Cases throughout the Pandemic with a Special Focus on Lung Manifestations and Extrapulmonary Vascular Abdominal Complications

**DOI:** 10.3390/biomedicines11082113

**Published:** 2023-07-26

**Authors:** Barbara Brogna, Elio Bignardi, Antonia Megliola, Antonietta Laporta, Andrea La Rocca, Mena Volpe, Lanfranco Aquilino Musto

**Affiliations:** 1Department of Interventional and Emergency Radiology, San Giuseppe Moscati Hospital, 83100 Avellino, Italy; antoniettalaporta6@gmail.com (A.L.); andrealarocca1989@libero.it (A.L.R.); mustolanfranco@gmail.com (L.A.M.); 2Department of Radiology, Francesco Ferrari Hospital, ASL Lecce, 73042 Casarano, Italy; eliobignardi60@gmail.com; 3Radiology Unit, “Frangipane” Hospital, ASL Avellino, 83031 Ariano Irpino, Italy; antoniameg@gmail.com (A.M.); menavolpe@hotmail.com (M.V.)

**Keywords:** typical COVID-19 pneumonia, atypical features, breakthrough infection, ARDS, barotrauma complications, pneumomediastinum, pneumothorax, lung thromboembolism, vascular disease, abdominal ischemic and hemorrhagic COVID-19 complications

## Abstract

With the Omicron wave, SARS-CoV-2 infections improved, with less lung involvement and few cases of severe manifestations. In this pictorial review, there is a summary of the pathogenesis with particular focus on the interaction of the immune system and gut and lung axis in both pulmonary and extrapulmonary manifestations of COVID-19 and the computed tomography (CT) imaging features of COVID-19 pneumonia from the beginning of the pandemic, describing the typical features of COVID-19 pneumonia following the Delta variant and the atypical features appearing during the Omicron wave. There is also an outline of the typical features of COVID-19 pneumonia in cases of breakthrough infection, including secondary lung complications such as acute respiratory distress disease (ARDS), pneumomediastinum, pneumothorax, and lung pulmonary thromboembolism, which were more frequent during the first waves of the pandemic. Finally, there is a description of vascular extrapulmonary complications, including both ischemic and hemorrhagic abdominal complications.

## 1. Introduction

At the end of 2019, a novel pneumonia of unknown origin was first detected in Wuhan, China. The virus responsible for this “unknown pneumonia” was called SARS-CoV-2, and the disease was subsequently named COVID-19. The first cases of this novel disease appeared in China. However, several authors documented the presence of antibodies for SARS-CoV-2 in the European population before its discovery in China [1,2,3]. COVID-19 shocked health and economic systems worldwide. Industrialized countries experienced more damage in terms of deaths and economic stress caused by applying measures of closures and isolation, mainly based on political decisions. COVID-19 attracted the attention of many researchers, and many publications were also produced by non-academics with important contributions. COVID-19′s origin is considered natural [4,5,6,7]. Nonetheless, several authors have not excluded the hypothesis of a laboratory origin, posing the question of biological risks [8,9,10,11]. The first phase of the pandemic was characterized by severe cases of interstitial pneumonia and lung thromboembolisms and a rapid evolution of acute respiratory distress disease (ARDS). Initially, autopsies on deceased COVID-19 patients were scarce due to significant concerns, such as the risk of contagion, a lack of healthcare specialists, and the abrupt outbreak of the disease [12]. However, with increased clinical experience and the first COVID-19 autopsies, the crucial role of vasculature damage and the hyperinflammatory state in COVID-19 pathology became clear [12,13]. Therefore, COVID-19 evidently involves not only the lungs but also the gastrointestinal and central nervous systems [14,15]. The gastrointestinal system’s role remains under investigation, with the gut microbiome possibly implicated in disease severity [16,17]. Radiologists played an important role during the emergence of the pandemic. At the beginning of the pandemic, Chinese radiologists reported in detail the characteristics of COVID-19 pneumonia, and the main features of this disease were easily available worldwide [18,19]. Chest CT was used as the main imaging modality to score interstitial pneumonia and, along with clinical parameters such as age, D-dimer level, and albumin. It was an important predictive and prognostic indicator for disease severity and progression [20,21,22]. CT also depicted the systemic complications of the disease caused by an exaggerated inflammatory response, which involved severe alveolar damage in the lungs and exacerbated the hypercoagulation that led to venous thrombosis, ischemic attack, vascular dysfunction, and the infarction of the visceral abdominal organs [23,24]. Some hemorrhagic complications in the first waves of the pandemic were also a consequence of anticoagulant therapy [25,26]. However, the disease presentations improved with the Omicron variant, with less lung involvement and few cases of severe manifestations [27]. Vaccination and acquired immunity together with the lower virulence of the Omicron variant decreased the rate of severe infections [28]. Nevertheless, the current COVID-19 vaccines are limited, presenting a short duration of protection and risk of breakthrough infection, i.e., COVID-19 infection in a fully vaccinated person [29,30,31,32,33]. Risk factors for pneumonia severity in fully vaccinated people include old age, an immunosuppressed status, and the presence of comorbidities [29,30,31,32,33,34]. This paper presents a pictorial review on the pathogenesis, also focusing on the role of the gut and lung axis, the main features of radiological presentations of COVID-19 throughout the pandemic, with a focus on secondary lung complications, lung features in breakthrough infections, and abdominal vascular complications.

## 2. SARS-CoV-2 Pathogenesis

### The Important Role of the Immune System and of Gut–Lung Axis in Both Pulmonary and Extrapulmonary Manifestations

SARS-CoV-2 was a previously unknown coronavirus, and the first viral genome was sequenced using high-throughput sequencing (HTS) from a sample collected in Wuhan, China [35]. SARS-CoV-2 shows high genetic similarities with SARS-CoV, the causative agent of SARS, and as SARS-CoV, it binds through the spike protein to the angiotensin-converting enzyme 2 (ACE2) receptors, which are largely expressed in the lung, heart, vascular, and gastrointestinal systems [36,37]. ACE2 expression usually increases with age, and this feature could be a possible explanation for the aggressive form of COVID-19 disease that was more commonly found in the elderly [37]. 

COVID-19 manifestations can vary from an asymptomatic to a mild symptomatic disease, with a prevalence of respiratory symptoms such as cough and dyspnea to moderate/severe disease with lung interstitial pneumonia to acute distress respiratory syndromes to multisystemic involvement [38,39]. However, COVID-19 can manifest itself with extrapulmonary symptoms as well, together with gastrointestinal and neurological manifestations [38,39].

Throughout the pandemic, it became clear that a dysregulation of the immune response, also known as the “cytokine storms”, played a key role in disease severity, both in pulmonary and in extrapulmonary manifestations [40,41]. In results from some recent studies regarding disease severity, there seems to be an involvement of the intricate complex of both the innate and adaptive immune response, also evidencing an imbalance between the hyperinflammatory and the immunosuppressive phenomena [40,41,42,43]. On the other hand, the disease severity seems to be directly correlated with early humoral immune response. It also seems inversely related to early cellular immune response [40,41,42]. Moreover, dysregulated innate and adaptive immune responses usually occur in parallel with activation of coagulation, and these features could explain both the vascular pulmonary and extrapulmonary manifestations of COVID-19 [40,41,42,43,44].

However, the interaction between the immunity and inflammations of the lung and gut microbiome and the role of gut–lung axis in the immune response in the context of COVID-19 continues to be studied [45,46,47]. Observing from a different perspective, it is evident that an alteration of the gut microbiome can be associated with many respiratory diseases. Furthermore, dysbiosis of gut microbiota was found to be implicated in various alterations of the immune system [46,47]. The intestinal microbiota affects the expression of the type I interferon receptors in respiratory epithelial cells [47].

During the pandemic, cases of SARS-CoV-2 positivity in the stool also emerged, and this feature posed some questions on SARS-CoV-2 epidemiology [48,49,50]. 

Recently, researchers have found a close interaction between SARS-CoV-2 and intestinal bacteria, reporting some toxicological interactions between the bacteria of the microbiome and SARS-CoV-2. This discovery could explain the lung involvement severity and could have a role in the long-COVID symptoms [16,50,51,52,53]. In particular, some “toxin-like” peptides were observed in bacteria stools samples of COVID-19 patients, and the toxins found are similar to conotoxin-like peptides, phospholipase A2, and neurotoxins. These findings could also explain the atypical features of ARDS in COVID-19 patients [13].

With this consideration, this scenario could open new therapeutic approaches also for the development of new vaccines.

## 3. Lung Complications 

### 3.1. COVID-19 Pneumonia: Role of Chest CT and Chest Features from the Wild-Type to the Omicron Variant

The lungs were frequently involved in SARS-CoV-2 infection during the first pandemic’s waves; about 80% of patients with COVID-19 developed pneumonia up to and including the Delta variant [54]. 

Chest CT was widely used in China and globally in the early phases of the pandemic before the results of molecular SARS-CoV-2 nasal swabs were rapid and available. Chest CT was used due to its rapidity and to limit the contact between the radiographers and the infected patients [21,37,55,56].

Chest CT was also important in the staging and follow-up of COVID-19 interstitial pneumonia [21,55,56,57] (Figure 1). 

However, the Fleishner Society recommended the use of chest CT in emergency settings only for the triage of symptomatic COVID-19 patients presenting respiratory symptoms with a decline in O_2_ saturation level and patients at major risk of complications; it is not indicated to screen patients with COVID-19 [55]. At the beginning of the pandemic, Chinese radiologists described in detail the typical distribution of COVID-19 pneumonia, i.e., peripheral or peripheral and central. Ground glass opacities (GGOs) were visible in the early phase; as the pathology progressed, GGOs with a crazy-paving appearance and consolidations were evident along with subpleural and parenchymal bands. There was also a predominance of architectural distortion; in the peak stage, these findings had progressed, while the later stage demonstrated their resolution [55,56,57,58,59,60] (Figure 2).

Other typical signs were vessel enlargements, the reversed halo sign, and subpleural curvilinear lines (Figure 1 and Table 1). 

Atypical appearances included lobar consolidation, lung nodules or masses, miliary patterns, cavitation, and pleural effusion [61]. 

The differential diagnosis of COVID-19 pneumonia includes other infective forms of pneumonia such as bacterial pneumonia (usually lobar consolidation) and atypical pneumonia (multifocal distribution with centrilobular nodules), other viral forms of pneumonia, and cardiovascular pathologies such as pulmonary edema [62] (Figure 3 and Table 1).

It is usually possible to differentiate COVID-19 pneumonia from other viral pneumonia, such as that of influenza or syncytial and parainfluenza viruses and other coronaviruses, with a CT scan [63,64,65]. The diagnostic accuracy usually increases with the use of deep learning [66]. The resulting CT features that have been reported as the most strongly associated with COVID-19 pneumonia are the subpleural band of GGOs, where thickening of the bronchial walls and micronodules are usually more frequently associated with other viral pneumonia [63,64,65] (Figure 3c). However, some CT findings may overlap between COVID-19 pneumonia in influenza pneumonia [65]. Both COVID-19 and influenza pneumonia could show a diffused distribution; however, a lower-lobe-predominant distribution was reported to be more common in COVID-19, and an upper-lung-predominant distribution is slightly more common in influenza [65]. 

The radiology reporting system during the pandemic included the typical, atypical, and indeterminate features of COVID-19 pneumonia on the basis of RSNA classification; the grading of suspicion (using the CO-RADS classification); and the staging of pneumonia extension based on chest CT severity score (CT-SS) with semiquantitative or quantitative methods [37]. Typical features of COVID-19 pneumonia have been reported since the Delta variant [54,67,68,69,70,71]. Blanca et al. [70] found no significant differences in terms of mortality between the first and second waves of the pandemic. However, a worse long-term prognosis was found for the second wave [72] (Figure 4). These findings could be explained by the fact that the patients infected in the first wave were significantly younger than those in the second wave [72,73,74]. 

Inui et al. [68] compared chest CT features and severity scores between patients infected with wild-type, Alpha, and Delta variants of SARS-CoV-2 and found that pneumonia severity was highest for the Delta variant. COVID-19 pneumonia caused by the Delta variant also showed GGOs with reticulation or consolidation and more rapid repair capabilities than the wild-type and Alpha variants (Figure 5 and Figure 6). In contrast, no differences were found in the CT results between the wild-type and Alpha variants [67]. 

These features could be caused by the pathogenicity of the Delta variant, which shows mutations in the spike protein that enhance fusogenicity and facilitate more efficient expansion through the human body via cell–cell fusion than the wild-type and Alpha variants [67]. However, Han et al. [75] showed a decrease in the severity of pneumonia from the original strain to the Delta and Omicron variants.

The most recent variant, Omicron, despite being the most highly mutated and presenting increased contagiousness and infectivity, tends to involve the lungs to a lesser extent and predominantly infects the upper respiratory tract [54,68,74,75,76,77,78]. These findings suggest that the genetic and pathogenic characteristics of SARS-CoV-2 have evolved over time. In fact, the Omicron variant replicates in the bronchial epithelium cells, whereas previous waves of SARS-CoV-2 replicated in the alveolar epithelium [54,77].

Patients with the Omicron variant presented a lower pneumonia prevalence than in the previous waves and a peribronchial distribution (rather than peripheral), as compared to patients infected with the Delta variant [27,28,54,75,76,77,78] (Table 2).

The Omicron variant has been reported to manifest itself with lung involvement in 34% of cases [54]. However, lung involvement in the Omicron variant has been more frequently characterized by atypical chest CT findings such as cluster-like GGOs, randomly distributed GGOs, or the absence of pneumonia [54,68,69,74,75,76,77,78] (Figure 7, Figure 8, Figure 9 and Figure 10). Other atypical findings in patients infected with the Omicron variant included centrilobular nodules or tree-in-bud opacities [75,76].

Askani et al. [67] found that the Delta variant was more commonly associated with the typical signs of COVID-19 pneumonia than the Omicron variant and showed vacuolar CT signs more frequently. 

Patients with pneumonia during the Omicron wave tended to be older than those without COVID-19 pneumonia [27,28,54,77].

However, the persistence of fever during the Omicron wave could also be associated with lung involvement in young patients without comorbidities. 

### 3.2. COVID-19 Pneumonia after Breakthrough Infections

COVID-19 vaccines were created using various technologies: mRNA, protein subunits, inactivated non-replicating viral vectors, and DNA [79].

These vaccines proved to be highly effective in reducing the rates of hospitalization and severe presentations in patients affected by SARS-CoV-2 [79,80,81]. 

However, the progressive waning of antibody levels over time, together with the emergence of new viral variants, led to an increased risk of breakthrough infections [37,82,83,84].

A breakthrough infection is defined as a COVID-19 infection occurring ≥14 days after a full dose of a COVID-19 vaccine. Breakthrough infections are common, even after a third dose, and effectiveness against symptomatic disease appears to decrease 12–16 weeks after vaccination [84,85,86].

Several studies have reported that COVID-19 breakthrough infections usually appear with mild symptoms, and vaccination reduces pneumonia severity [87,88,89]. The imaging findings for breakthrough infections with pneumonia are similar to those for unvaccinated patients, though usually less severe; nevertheless, the severe forms are also possible in fully vaccinated patients with risk factors such as old age, immunosuppressed status, comorbidities, and anti-IFN-α autoantibodies [30,31,32,33,34,89,90,91,92] (Figure 11).

Old age can be considered an independent risk factor for pneumonia severity in both the unvaccinated population and breakthrough infections [31,32,77]. A lower antibody response after vaccination has been reported in older individuals [31,32]. Mirouse et al. [31] found that old age and disease severity were independently associated with high mortality. Tong et al. [32] determined that old age was the only independent risk factor for pneumonia development in patients infected with the Omicron variant (Figure 12 and Figure 13). 

Ben et al. [91] reported the prevalence of severe or critical disease in vaccinated COVID-19 patients in 10.8% of cases. Granata et al. [92] found that the risk for severe COVID-19 outcomes after primary vaccination was higher among persons aged ≥65 years suffering from immunosuppression; diabetes; and chronic kidney, cardiac, pulmonary, neurologic, or liver disease.

The risk of developing pneumonia after breakthrough infection increase is higher in patients with cancer. The risk is heightened for those receiving active chemotherapy and anti-CD20 therapy [30,82,93].

Virus variants increased the risk of breakthrough infections [82,87,88]. However, severe cases of COVID-19 pneumonia after breakthrough infections were prevalent in the Delta variant wave [69,75,77,93,94,95]. Lamacchia et al. [90] found that vaccinated and hospitalized COVID-19 patients were most likely infected with the Delta variant. The Delta variant caused more severe disease and was less susceptible to vaccines than previous lineages [37] (Figure 14). 

Nevertheless, several studies showed that pneumonia severity was lower in vaccinated than in unvaccinated patients [37,87,88,89,90,91,96,97]. Lee et al. [96] reported that the proportion of patients with negative CT scans was significantly greater in the fully vaccinated group than in the unvaccinated group. Verma et al. [97] reported that the mean CT severity score was significantly lower in completely vaccinated patients in comparison to incompletely vaccinated and non-vaccinated patients. Wada et al. [89] found that vaccinated patients, with or without a booster/additional vaccination, presented milder COVID-19 pneumonia on CT scans than unvaccinated patients during the period of the Delta and Omicron variants. Nevertheless, Granata et al. [92] found no difference between vaccinated and unvaccinated patients admitted to the ICU. However, vaccinated patients with pneumonia were usually older than unvaccinated patients and had comorbidities [98]. 

Nevertheless, regardless of vaccination status, pneumonia prevalence progressively declined during the Omicron wave [77]. The CT features in breakthrough infections and the CT features of COVID-19 pneumonia in the different waves of pandemic are summarized in Table 3. 

### 3.3. Secondary Lung Complications 

#### 3.3.1. Acute Respiratory Distress Syndrome

An uncontrolled immune response to SARS-CoV-2 infection can lead to acute respiratory distress syndrome (ARDS), which was a relatively common complication of COVID-19 pneumonia in the first waves [99,100,101,102,103,104]. COVID-19 patients with ARDS were usually older and had comorbidities [100]. The majority of patients with ARDS required mechanical ventilation and were prone to complications including barotrauma with alveolar rupture and superimposed bacterial pneumonia [101,102,103]. 

COVID-19 ARDS is caused by both direct viral action and the indirect effect of the hyperactivation of the host immune system, which can cause the excessive production of proinflammatory cytokines, i.e., cytokine storms. COVID-19 ARDS differs from typical ARDS due to the existence of endothelial injury characterized by the presence of pulmonary capillary microthrombi, in addition to typical diffused alveolar injury [99,100,101,102,103,104,105]. This was revealed by histopathological examinations, although autopsies were rare in the first wave of the pandemic [103]. SARS-CoV-2 mainly attacks the endothelium through the ACE2 receptor. The injured pulmonary endothelium loses lung perfusion regulation with the activation of an intrapulmonary shunt. Thus, the lung volume and lung compliance are almost normal [99,100,101,102,103,104,105]. Therefore, COVID-19 ARDS is characterized by silent or happy hypoxemia. The imaging features of COVID-19 ARDS were similar to those of ARDS with other etiologies [103]. Chest CT findings were characterized by the anteroposterior gradient of the lung opacities. The rapid progression of lung opacities involving all five lobes in a patient with COVID-19 should also increase concern for ARDS. In the acute phase, a tendency towards dense consolidation involving the dependent posterior lower lobes while mostly sparing the anterior or non-dependent area has been observed [37,103] (Figure 15). 

ARDS in COVID-19 patients has been frequent since the Delta variant [37,99,104,105,106]. However, the risk of ARDS was lower during the Omicron variant [107]. Some rare cases of ARDS were reported after COVID-19 vaccinations as possible consequences of antibody-dependent enhancement (ADE) [108,109] (Figure 16).

#### 3.3.2. Pneumomediastinum and Pneumothorax

Pneumomediastinum (PMS) is the presence of air in the mediastinum. PMS has been identified as a complication in lung infections such as influenza and previous coronavirus infections such as SARS-CoV-1 and Middle East respiratory syndrome (MERS). It appears to be a marker of severe COVID-19 pneumonitis [110,111]. This condition can be spontaneous or secondary. The secondary form is the most common, with high incidence during the first three waves of the pandemic in critically ill patients (Figure 15) and may be related to mechanical ventilation in patients with ARDS receiving high positive pressure ventilation (PPV) [110,111,112,113,114,115,116]. Pneumothorax (PNX) was found to co-exist with PMS in 40.3% of cases [110]. Several studies have reported that the incidence of barotrauma in COVID-19 patients was 14% [114,115]. However, Martinelli et al. [117] reported a much higher incidence placing it at 25%. Subcutaneous emphysema is usually associated with PMS and PTX. The spontaneous forms of PMS and PNX have no apparent causes; however, several predisposing and precipitating factors have been identified, such as asthma, respiratory infections, lung cysts, inhaled drug use, corticosteroids, and the inhalation of irritants, as well as several anatomical predisposing alterations, including tracheomalacia [111,118,119,120,121,122]. Lung damage with alveolar rupture in COVID-19 infections can probably lead to interstitial pneumothorax, which can cause spontaneous PMS related to the Macklin effect [110,111,118,119,120,121,122]. In severe cases of COVID-19 pneumonia, parenchymal destruction can lead to the formation of cavitation and lung cysts, which can cause PTX and PMS after rupture [111,118,119,120,121,122]. 

The spontaneous forms of PTX and PMS occurred in 1–2% of all COVID-19 pneumonia patients [118,122]. In rare cases, spontaneous PMS was associated with PTX [123]. The prevalence of these conditions increased during the second wave of the pandemic with the increasing use of oxygen through non-invasive ventilation and steroid therapy [118,121,122,123,124,125,126]. The high airway pressures delivered by these modalities of respiration probably supported the spontaneous rupture of fragile small airways infected by the virus (Figure 17 and Figure 18). 

However, Palumbo et al. [126] hypothesized that the increased incidence of SMPS and SPTX during the second pandemic wave was caused by the increased use of dexamethasone, which might have induced lung frailty. Rare cases of SPMS were usually reported during the Omicron wave [127].

#### 3.3.3. Pulmonary Fibrosis 

Approximately one-third of patients hospitalized with COVID-19 pneumonia presented abnormalities on chest CT scans 1 year from infection [128,129]. In patients with severe disease, CT studies after hospital discharge reported fibrotic-like changes such as COVID-19 sequelae (from 3 to 12 months following COVID-19 pneumonia). Furthermore, parenchymal bands, interlobular septal thickening, coarse reticulations, and bronchiectasis are the most frequently reported CT findings (Figure 19). Air trapping has also been reported as a long-term finding in COVID-19 survivors in several studies [128]. However, the term “fibrosis” is problematic for CT because fibrosis tissues are not confirmed, and real fibrosis is usually an irreversible condition [128]. Therefore, it would be more correct when describing these observations as being fibrotic-like changes that could be caused by acute inflammatory damage, which usually resolves over time. Several studies have shown that in most patients, pulmonary post-COVID-19 fibrosis-like changes resolved over time [129,130,131]. Bocchino et al. [131] reported that at 1 year, lung abnormalities due to COVID-19 pneumonia were completely resolved in 93% of patients. However, only a minority of patients showed persistent fibrotic changes via CT, especially those with moderate-to-severed disease who required intensive care therapy or mechanical ventilation [128,129,130,131,132]. Older patients with severe comorbidities were also more prone to develop fibrotic changes [132]. 

#### 3.3.4. Pulmonary Thromboembolism

In the early phase of the pandemic, the important role of a prothrombotic state in the virus infection became rapidly clear, with evidence of dilated pulmonary vasculature in regions affected by pneumonia according to CT and a high incidence of pulmonary thromboembolism (PE), according to CT pulmonary angiography (CTPA) [12,133,134,135,136]. Nevertheless, during the first phase of the pandemic, complete autopsy studies were rarely performed because of the risk of infection, and PE could have been underestimated [133,134,135]. However, the first autopsies highlighted the importance of thromboembolic events in COVID-19, suggesting de novo coagulopathy in patients with COVID-19, and autopsy studies reported the presence of microscopic thrombi formation in the lung histology [12,133,134,135,136]. Several authors found a higher incidence of pulmonary embolism with or without deep venous thrombosis in COVID-19 patients with no history of venous thromboembolism (VTE). Thrombotic events in hospitalized COVID-19 patients were observed despite VTE prophylaxis [10,12,136,137,138,139]. A recent meta-analysis reported that autoptic findings of acute PE in COVID-19 patients were present in about 30% of the subjects [10]. Radiology case reviews reported high rates of PE, ranging from 17% and 50%, especially in patients who were admitted to the ICU [133] (Figure 20, Figure 21 and Figure 22). 

This increased prevalence occurred despite a simultaneous decrease in the overall number of CTPA exams performed during the pandemic. Recently, Wusmuller et al. [133] showed that the prevalence of PE increased significantly as reported by CTPA exams during the first wave of the COVID-19 pandemic, despite the reduction in the number of radiological examinations carried out due to lockdown measures. Segmental arteries were generally the most common location for PE [24,140] (Figure 23). 

Katsoularis et al. [141] found that the risk of a first pulmonary embolism event was highest during the first pandemic wave when compared to the second and third waves. However, Manzur-Pineda et al. [142] determined that 28% of patients showed PE during the Delta variant peak. 

Nevertheless, the rate of thrombotic complications reduced during the Omicron wave [140,141,142,143,144]. The incidence of PE also showed a progressive reduction during the Delta variant wave compared to the ancestral variants [140,141]. 

However, a prothrombotic state associated with COVID-19 has been reported even in subclinical infections [142,143,144].

An increased risk of thrombosis was also found in the early and late sub-acute phases [143,144]. Patients with moderate and severe disease are reportedly at an increased risk of developing chronic thromboembolic pulmonary hypertension [145,146]. All the lung complications are summarized in Table 3.

## 4. Vascular Abdominal Extrapulmonary Complications

SARS-CoV-2 infection can cause a variety of extrapulmonary complications that are usually secondary to the state of hypercoagulability caused directly by the viral invasion of the endothelial cells and indirectly by the activation of the immune system with a high concentration of circulating proinflammatory cytokines [147,148,149,150]. Ischemic and hemorrhagic abdominal complications were mainly reported in the first two waves of the pandemic [147,149,150,151,152,153,154]. Some case series were also reported during the Delta and Omicron waves [155,156]. Both thrombotic and hemorrhagic abdominal complications were particularly associated with extended lung disease, especially in ICU patients and the elderly [147,149,150,157]. Other reported risk factors for developing these complications were arterial hypertension, diabetes, and obesity [147,150,157]. Hemorrhagic complications were also reported as a possible complication of the anticoagulant treatment usually applied for the management and prevention of PE [147,152,158]. Corticosteroid therapy was also associated with a slight increase in the incidence of gastrointestinal bleeding [158]. However, hemorrhagic abdominal complications were found to be a consequence of COVID-19-associated coagulopathies [153]. Contrast-enhanced CT (CECT) with multiphasic acquisition played a key role in the diagnosis of these complications and provided clinicians with important information for patient management. CECT is usually considered the imaging method of choice for evaluating patients with a clinical suspicion of arterial and venous thrombosis because it allows the direct visualization of the thrombus [159]. Additionally, CECT with angiographic acquisition is very useful for locating the source of active bleeding. Ischemic abdominal complications in COVID-19 patients affected the small and large bowel, the liver, the spleen, and the renal parenchyma [24,157,159]. ACE2 receptors are highly expressed on endothelial cells in the intestine and kidney [24,148,156,159]. 

### 4.1. Small Bowel Ischemia and Ischemic Colitis

Small bowel ischemia was the most frequent abdominal CT finding in COVID-19 patients with intestinal ischemia, followed by ischemic colitis. These complications were found more frequently in patients with severe disease [147,149,150,159,160,161,162,163]. Non-occlusive mesenteric ischemia (NOMI) was also a common pattern of bowel involvement, indicating microvascular involvement [161]. Small bowel ischemia had a higher mortality rate, and radiologists played a key role in diagnosis because a prompt diagnosis could prevent mortality in these patients [150,159,160,161,162,163,164]. Most COVID-19 patients with small bowel ischemia had a fatal outcome [147,149,159,160,161,162,163,164]. The CT pattern of small bowel ischemia varied based on vascular involvement (arterial or venous occlusion and hypoperfusion) and the timing of occlusion or hypoperfusion [159,160,161,162,163,164]. The early phase of arterial occlusion is characterized by a spastic reflex ileus with a diminished bowel mural enhancement [162,164]. CECT can show the presence of emboli or thrombi as a filling defect in the lumen of the artery. However, in some cases, if they are small and peripherally localized, their identification can be difficult. The mesentery is bloodless. In the later phase, the ischemia rapidly evolves into infarction with progression from a hypotonic reflex ileus to a paralytic ileus due to wall necrosis with perforation [159,160,161,162,163,164]. Small bowel venous ischemia is usually characterized in CT by the target appearance of the ischemic bowel with an inner hyperdense ring due to mucosal hypervascularity, hemorrhage, and ulceration; a middle hypodense edematous submucosa; and a normal or slightly thickened muscularis propria. If the vascular impairment persists, the condition progresses to intestinal infarction. In the early phase of NOMI, the intestinal wall injuries are usually similar to those in the ischemic phase [160,162]. If no reperfusion occurs or the collateral circulation is ineffective, no enhancement is observed, with wall thinning due to the necrosis. The colon, not only the intestinal wall, is also involved in hypoperfusion injury. However, if reperfusion occurs, it is possible to visualize via CT the target appearance of the intestinal and colic wall [162]. Ischemic colitis is more frequently non-occlusive in nature with NOMI CT patterns [159,160,161,162,163,164].

Bonaffini et al. [147] reported 10 cases of ischemic abdominal complications in the first wave of the pandemic, and seven of these received anticoagulant therapy. Six cases were related to small bowel ischemia and four to ischemic colitis. Sardeck et al. [156] reported 21 cases of macroscopic intestinal ischemia, which appeared about 7 days after COVID-19 diagnosis with a range of 2–21 days. Most of the patients were male and had a mean age of 61.5 years. In this case series, no patients received anticoagulant treatment before hospital admission, and only three patients had received a booster dose of COVID-19 vaccines, with one patient having received two doses. Almost half of the patients with small bowel ischemia in this study died. However, in most of the published cases of small bowel and colic ischemia, no evidence of arterial or venous acute thrombosis was found, suggesting the important role of microvascular obstruction in these patients [147,149,157,160,165,166]. In the recent review of Oja et al. [160], the thrombosis of the SMA was found in 24.9% of cases. Ischemic colitis in COVID-19 patients is relatively rare and occurs as a result of colonic hypoperfusion secondary to the hemodynamic compromise in severe cases of COVID-19 infection [165,166,167] (Figure 24). The left colon is most commonly affected and includes the watershed areas from the splenic flexure to the sigmoid colon. Rare cases have also been reported in the right colon, probably due to microvascular thrombosis [168]. Additionally, rare cases of ischemic colitis have been reported after COVID-19 vaccination [169,170]. 

### 4.2. Splenic and Renal Infarction

Splenic and renal infarction are rare ischemic abdominal complications in COVID-19 patients and have only been presented in case reports and a few case series [147,149,171,172,173,174,175,176,177,178,179]. Splenic and renal infarctions generally have a minor clinical impact compared to intestinal infarction. In very few cases, associations between these complications have been reported [147,180,181,182,183], and in some rare cases these were the first manifestations of a COVID-19 infection [171,177]. These complications are usually secondary to the state of hypercoagulability in COVID-19 infection that leads to an increased risk of arterial and venous thrombosis and thromboembolic events (Figure 24 and Figure 25). 

Splenic infarctions follow arterial or venous occlusions, causing tissue ischemia and necrosis. Some rare cases have involved splenic infarction in COVID-19 patients without pulmonary involvement [171,173]. Childer et al. [174] described a case of splenic infarction in a COVID-19 patient associated with a floating thrombus in the aorta, suggesting a thromboembolic origin. Splenic infarction is also the most common finding associated with mesenteric ischemia in COVID-19 patients, followed by renal infarction [147,149]. In CECT, splenic infarction appears as a peripheral, wedge-shaped, hypo-enhancing region.

The management of splenic infarction is usually conservative. However, Dimitriou et al. [184] described a case of splenic infarction in an unvaccinated COVID-19 patient who required surgical treatment. A case of splenic infarction was also reported during the peak of the Delta variant wave in an unvaccinated male [185]. Rare cases of splenic infarction have been described after COVID-19 m-RNA vaccines [186,187,188,189,190] (Figure 26). 

The pathogenesis of renal infarction in COVID-19 is multifactorial and caused directly by the virus acting on the renal endothelium and indirectly by the cytokine storm, state of hypercoagulability, hemodynamic alterations, and glomerulopathy collapse [189]. Because the manifestations of a renal infarction can be subclinical, its diagnosis can be incidental and underestimated [189]. Renal infarction appears in CECT as a wedge-shaped focal hypodensity. Vasquez Espinosa et al. [190] reported cases of renal infarction in a young woman who was fully vaccinated. Rare cases of renal infarction have been reported in asymptomatic COVID-19 infections [177].

### 4.3. Hemorrhagic Abdominal Complications

Hemorrhagic abdominal complications in COVID-19 patients can occur due to predisposing factors such as the use of anticoagulants, thrombocytopenia, and the consumption of coagulation factors [152,153,154]. 

Several case reports and case series have described abdominal spontaneous hematoma in COVID-19 patients [152,153,154]. Spontaneous bleeding can theoretically involve all anatomic abdominal structures such as muscles, parenchymas, the retroperitoneum, and the peritoneum. Spontaneous bleeding is the dominant adverse event of anticoagulant treatment and was more frequent during the first two the pandemic waves. The most frequent locations were the retroperitoneum and the muscular abdominal walls [152,153,154,191,192,193,194,195] (Figure 27). The muscles most commonly affected were the iliopsoas and the rectus abdominis [152,153,154,191,192,193,194,195].

In most cases, patients were older, receiving a prophylactic dose of subcutaneous enoxaparin, and showed moderate-to-severe COVID-19 pneumonia. Evrev et al. [152] reported 11 cases of spontaneous abdominal and gastrointestinal bleeding in hospitalized COVID-19 patients, with a male predominance and an average age of 74 years. In nine patients, the retroperitoneal and gastrointestinal bleeding complications followed anticoagulant treatment and were associated with pneumonia severity. Rare cases of gastrointestinal bleeding were also reported during the Omicron wave and were related to the colon [156]. All vascular abdominal complications are summarized in Table 3.

## 5. Conclusions

In this pictorial review, the main features of COVID-19 pneumonia throughout the pandemic waves of the pandemic are summarized, together with the secondary lung complications and abdominal vascular complications. Radiologists played a key role in the first two pandemic waves when the ancestral variants showed higher pneumonia scores according to CT. Radiologists also played an important role in the diagnosis of COVID-19 pneumonia after breakthrough infections. CECT had a crucial role in the diagnosis of vascular abdominal complications, which were more frequent during the first two pandemic waves. The role of the interaction of the gut/lung axis microbiome with both innate and adaptive immune response remains under investigations with a possible important role in the disease development and severity. This scenario could open the way for the development of new therapeutic strategies. 

## Figures and Tables

**Figure 1 biomedicines-11-02113-f001:**
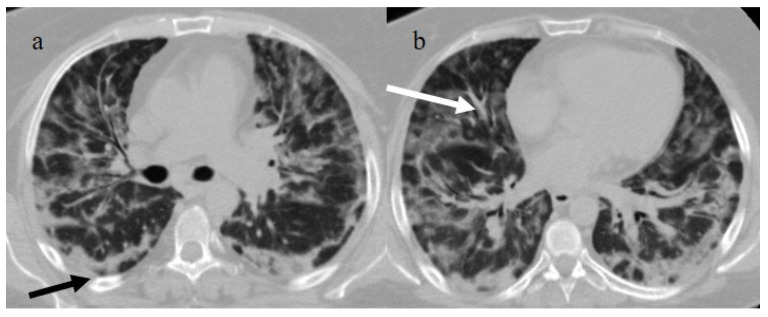
Typical central and peripheral distribution in COVID-19 at the beginning of the pandemic in Italy (March 2020) in a 50-year-old patient with a history of arterial hypertension. The CT was carried out in an emergency, with an initial diagnosis suspected COVID-19 pneumonia made by the radiologist before the RT-PCR results, and a suspicion of COVID-19 pneumonia was raised by radiologists. In the image, (**a**) the subpleural bands (black arrow) and (**b**) the vascular enlargement (white arrow) are visible.

**Figure 2 biomedicines-11-02113-f002:**
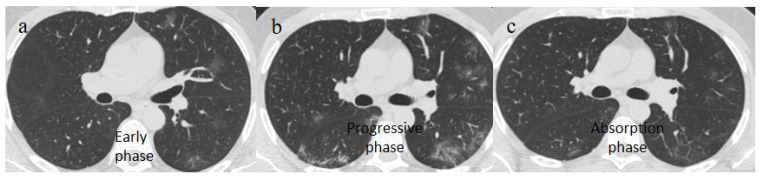
Chest CT COVID-19 pneumonia evolution during serial follow-up; (**a**) early phase with some GGO areas in a typical central and peripheral distribution; (**b**) the progressive phase with evolution in crazy paving with initial consolidation areas; (**c**) absorptive phase with reduction in previous inflammatory areas.

**Figure 3 biomedicines-11-02113-f003:**
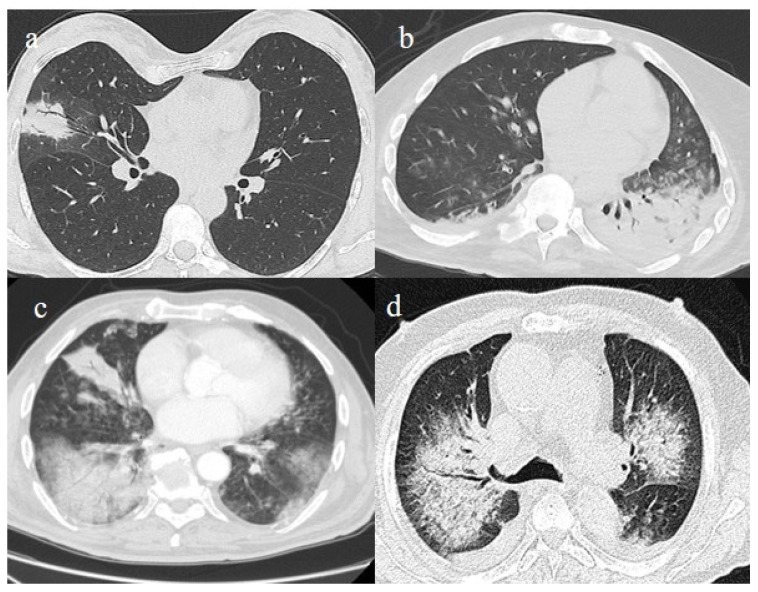
In this figure some differential diagnosis of COVID-19 pneumonia with atypical features are shown; in image (**a**), a solitary segmental bronchopneumonia is seen; image (**b**) shows an atypical bacteria pneumonia with consolidation areas and centrolobular nodules in a patient with immunosuppression state with positivity on BAL of Escherichia coli; image (**c**) illustrates an atypical pneumonia with multiple consolidations and centrolobular nodules in a patients with syncytial respiratory virus infection; image (**d**) shows consolidations with central hilar distribution ad bilateral pleural effusion with pulmonary edema.

**Figure 4 biomedicines-11-02113-f004:**
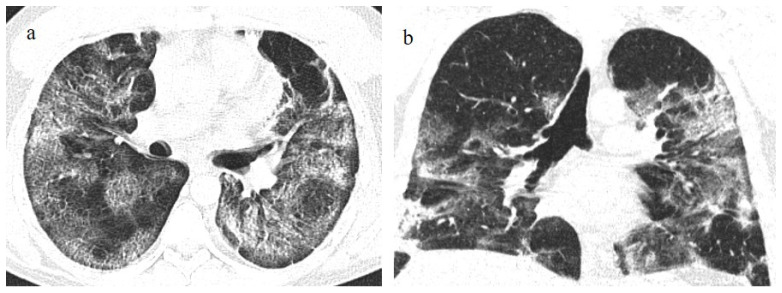
A 41-year-old patient during the Alpha wave (November 2020). The chest CT showed diffuse interstitial thickness with GGOs in a typical distribution on the axial plane (**a**) and coronal plane (**b**) and a high score. The patient had a history of bronchial asthma with any other known comorbidities. He was treated at home with paracetamol and azithromycin. He was hospitalized and died.

**Figure 5 biomedicines-11-02113-f005:**
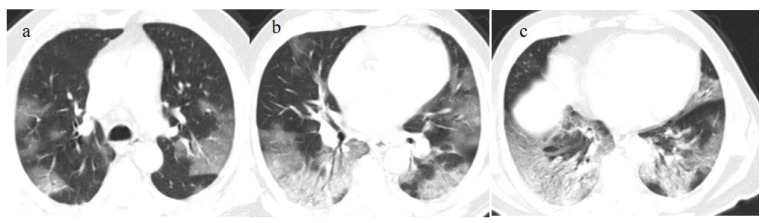
A 57-year-old unvaccinated patient during the Delta waves (in November 2021). On chest CT, he presented diffuse interstitial thickness with GGOs in a typical distribution in the superior and inferior lobes as visualized in image (**a**), in the middle and inferior lobes, in image (**b**) and at the lung base in image (**c**) and high score (CT-SS 14/25). The patient was hospitalized and died.

**Figure 6 biomedicines-11-02113-f006:**
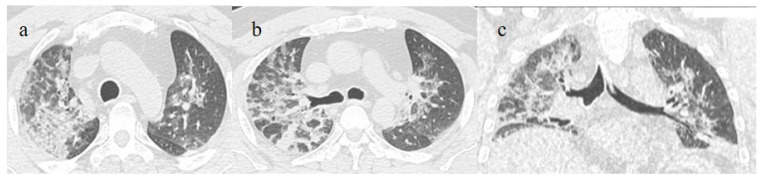
A 51-year-old man unvaccinated patient without known comorbidities during the Delta waves (in November 2021). On CT he presented diffuse interstitial thickness with consolidation areas in a typical distribution in the superior lobes and superior and inferior lobes, on axial plane, respectively in image (**a**), and in image (**b**); diffuse interstitial thickness and consolidations on coronal plane in image (**c**) and high score. The patient was hospitalized and survived.

**Figure 7 biomedicines-11-02113-f007:**
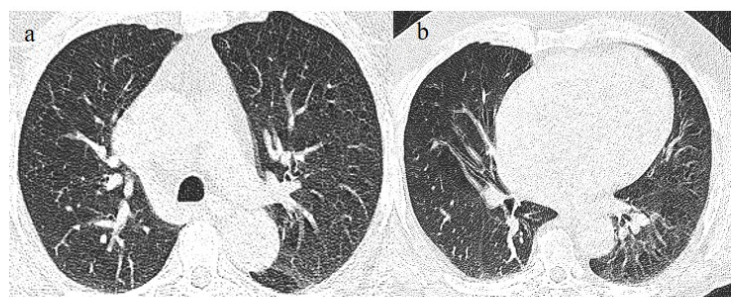
A 65-year-old patient in breakthrough infection with RT-PCR positivity for SARS-CoV-2 during the Omicron wave (in January 2023) with absence of pneumonia in chest CT as is depicted in image (**a**,**b**) on the axial plane.

**Figure 8 biomedicines-11-02113-f008:**
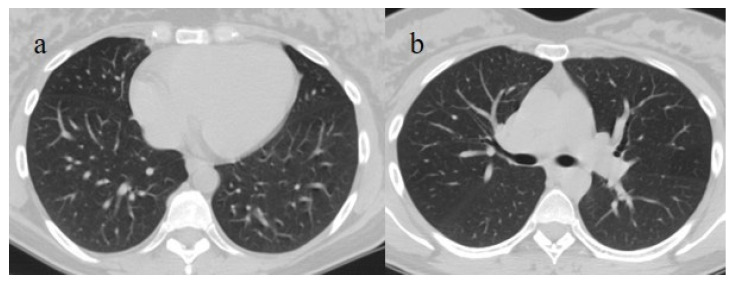
Absence of pneumonia in chest CT in images (**a**,**b**) in an unvaccinated 40-year-old patient with some comorbidities during the Omicron wave (February 2023).

**Figure 9 biomedicines-11-02113-f009:**
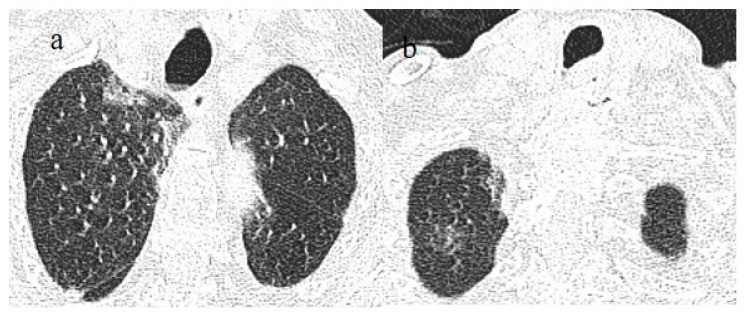
An 83-year-old patient with RT-PCR positivity for SARS-CoV-2 during the Omicron wave in breakthrough infection (in May 2023) with some cluster GGOs on the superior right lung lobe in image (**a**) and image (**b**). The patient was hospitalized and survived.

**Figure 10 biomedicines-11-02113-f010:**
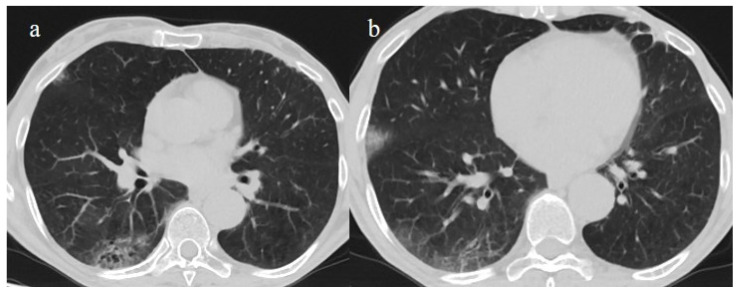
A 67-year-old unvaccinated patient with a history of hypertension, emphysema, and breast cancer and RT-PCR positivity for SARS-CoV-2 during the Omicron wave in February 2023 with some peripheral GGOs in the middle lobe, in image (**a**), and right inferior lobe, in image (**b**), with a peripheral and typical distribution. The patient was not hospitalized and did not have other complications.

**Figure 11 biomedicines-11-02113-f011:**
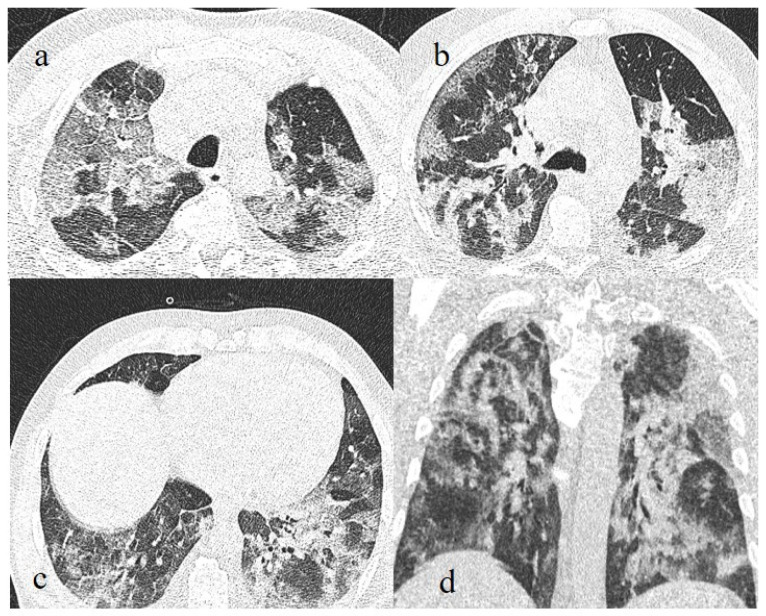
Breakthrough infection with severe pneumonia in a fully 61-year-old vaccinated patient during the transition period of the Alpha-Delta variant (June 2021), with a history of hypertension, ischemic heart disease, and a previous kidney transplant. In images (**a**–**c**), there is evidence of interstitial thickness with GGOs in the superior, superior and inferior, and inferior lobes, respectively, on the axial plane; the image (**d**) shows the multilobar distribution on the coronal plane.

**Figure 12 biomedicines-11-02113-f012:**
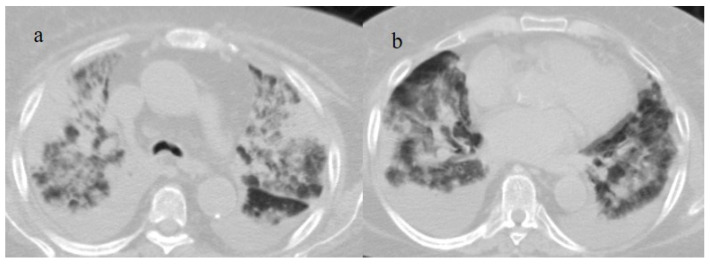
Breakthrough infection with severe pneumonia in an 80-year-old patient with a high CT score during the Omicron wave. The patient had a history of immunosuppression for renal transplant; images (**a**,**b**) show interstitial thickness with consolidations at the level of superior lobes (**a**) and inferior lobes on the axial plane; pleural effusion was also present. The patient was hospitalized and died.

**Figure 13 biomedicines-11-02113-f013:**
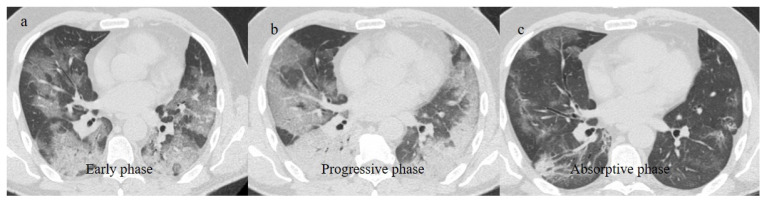
Chest CT scan in images (**a**–**c**) show the pneumonia evolution in a breakthrough infection in a 67-year-old patient during the Omicron wave (March 2023) with multiple comorbidities as hypertension and chronic heart failure.

**Figure 14 biomedicines-11-02113-f014:**
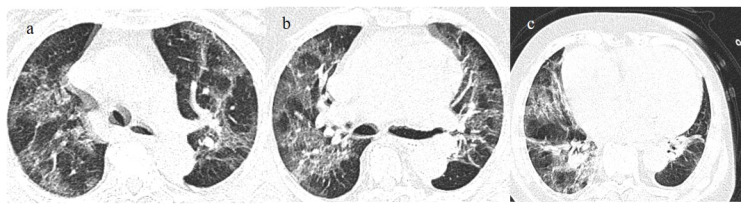
Breakthrough infection with a higher score on chest CT with a typical distribution during the Delta wave (November 2021). The images (**a**,**b**) show diffuse interstitial thickness with GGOs at the level of the superior lobes and inferior lobes on the axial plane; image (**c**) shows the middle and inferior lobes.

**Figure 15 biomedicines-11-02113-f015:**
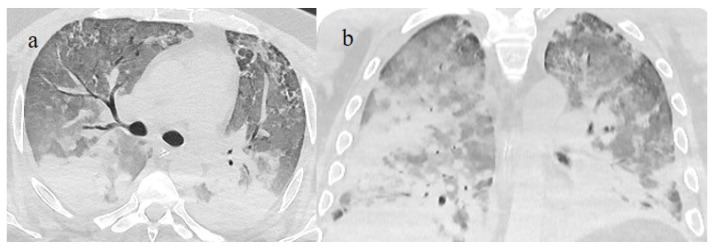
Chest CT scan of a 51-year-old patient during the Alpha variant period (February 2021) showing in image (**a**) posterior consolidations with diffuse GGOs compatible as ARDS on the axial plane and in image (**b**) on the coronal plane.

**Figure 16 biomedicines-11-02113-f016:**
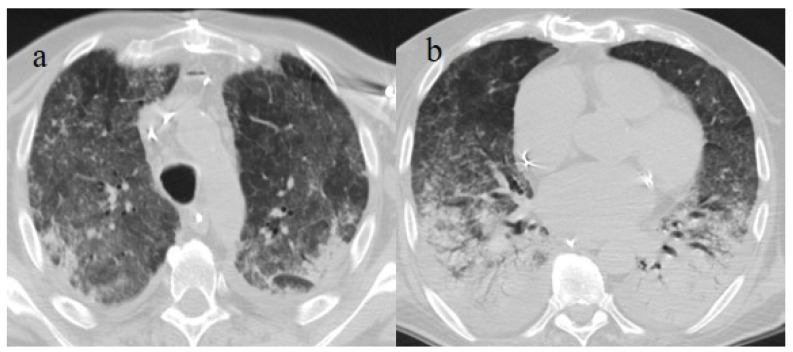
Chest CT scan of a 70-year-old patient during the Alpha variant period (April 2021) with ARDS findings in the superior lobes as shown in image (**a**) and inferior lobes in image (**b**). The patient developed severe COVID-19 pneumonia, which evolved into ARDS a few days after the COVID-19 vaccination.

**Figure 17 biomedicines-11-02113-f017:**
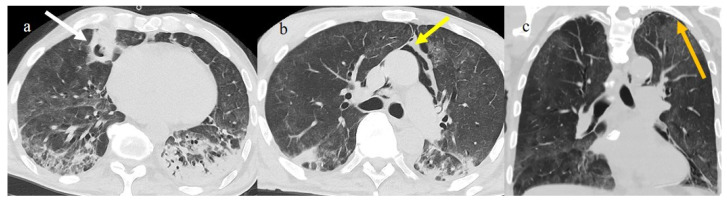
Chest CT during the second pandemic wave (November 2020) in a 71-year-old patient with a history of diabetes mellitus. The patient was under non-mechanical ventilation and developed a cavity lesion (white arrow) in image (**a**), pneumomediastinum (yellow arrow) in image (**b**), and a small pneumothorax (orange arrow) as visualized in image (**c**).

**Figure 18 biomedicines-11-02113-f018:**
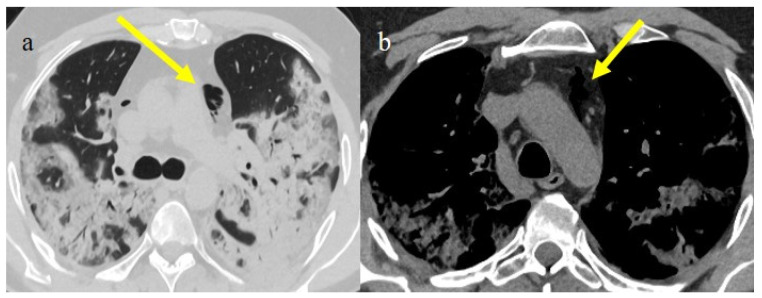
Chest CT during the second pandemic wave (November 2020) in a 40-year-old patient without known comorbidities. The patient was under non-mechanical ventilation with CPAP and developed pneumomediastinum (yellow arrow), as visualized on axial plane in image (**a**) (on lung window) and in image (**b**) mediastinal window; severe pneumonia with consolidations is also visualized in image (**a**).

**Figure 19 biomedicines-11-02113-f019:**
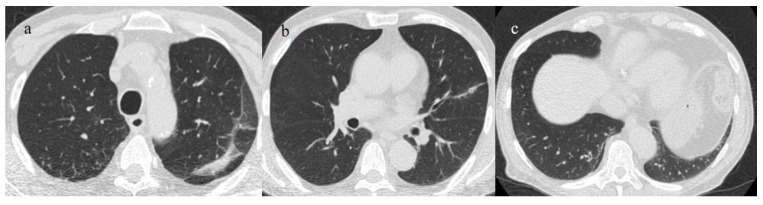
A 73-year-old patient with a history of hypertension and with some fibrotic-like change on the CT during the follow-up after 3 months of COVID-19 pneumonia, during the first pandemic wave (May 2020). There are some subpleural fibrotic bands in the apicodorsal segment of the left upper lobe (**a**), at the level of lingual segment (**b**), and in the inferior lobes (**c**).

**Figure 20 biomedicines-11-02113-f020:**
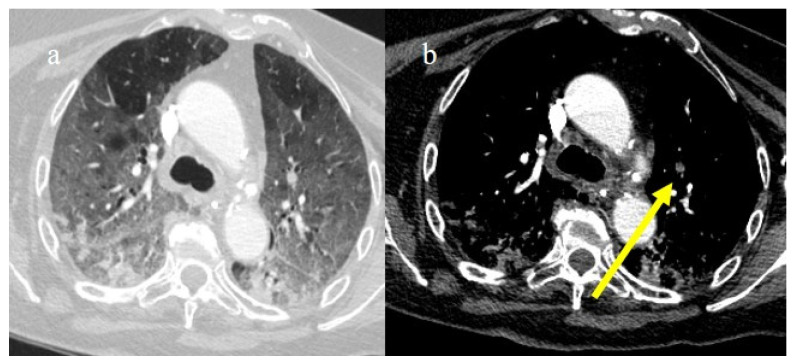
Chest CT of a 79-year-old patient in ICU during the second pandemic wave (October 2020) with severe COVID-19 pneumonia with CT-SS OF 20/25 with diffuse GGOs and initial consolidations as shown in image (**a**). On the CT pulmonary angiogram, a thrombosis was found in the anterior pulmonary segmentary artery of the left superior lobe, as shown in the image (**b**) (yellow arrow).

**Figure 21 biomedicines-11-02113-f021:**
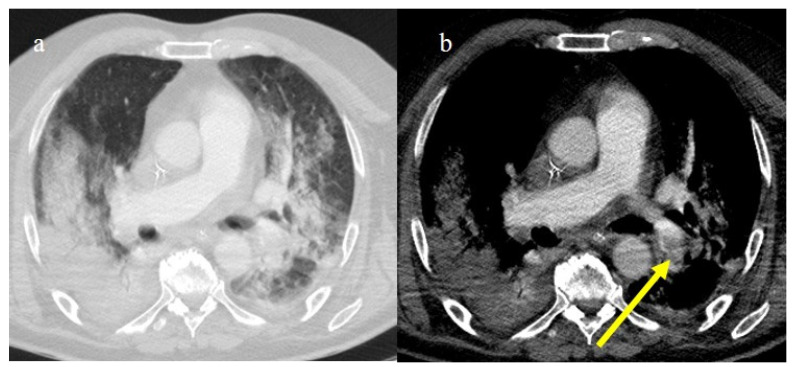
Chest CT of a 70-year-old patient in ICU in the second pandemic wave (November 2020) with severe COVID-19 pneumonia with CT-SS OF 23/25 with diffuse consolidations in the inferior lobe in image (**a**). The CT pulmonary angiogram showed that lung thromboembolism was present; in image (**b**), a thrombosis in the lobar artery branch for the left inferior lobe (yellow arrow) was described.

**Figure 22 biomedicines-11-02113-f022:**
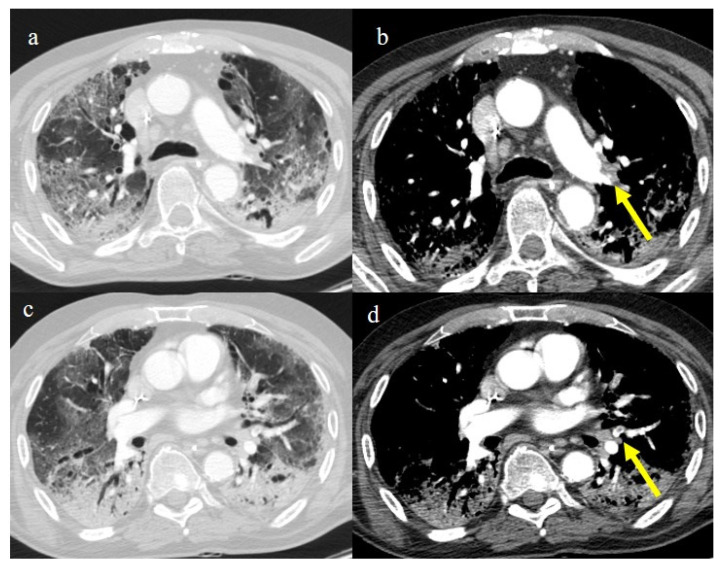
Chest CT of an 81-year-old patient in ICU during the transition period of Delta-Omicron variant (January 2022) (vaccination status unknown, probably concerns a breakthrough infection) with severe COVID-19 pneumonia with CT-SS of 23/25 with diffuse interstitial thickness and consolidations in the superior lobes, as visualized in image (**a**). The CT pulmonary angiogram showed thrombosis in the distal portion of the left pulmonary artery (yellow arrow) in image (**b**); severe pneumonia with consolidations in the apical segment of inferior lobes in image (**c**) and thrombosis in a segmentary branch (yellow arrow) in image (**d**).

**Figure 23 biomedicines-11-02113-f023:**
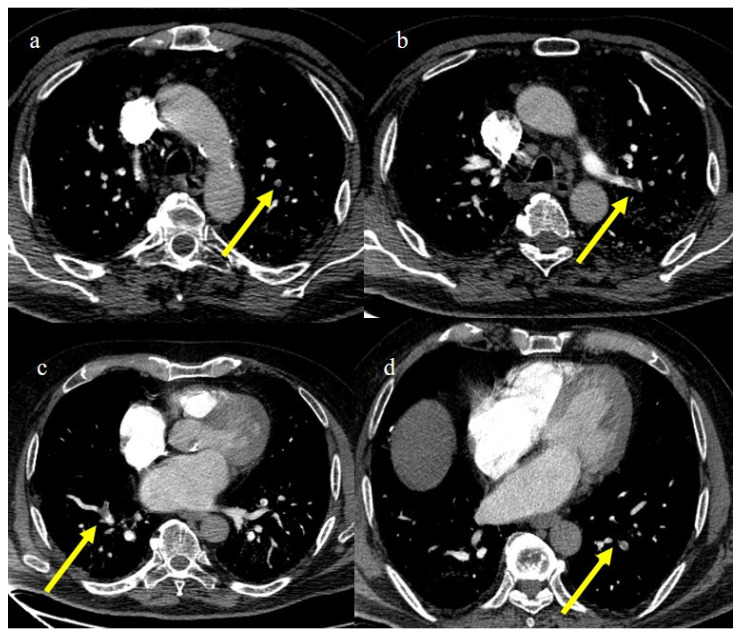
CT pulmonary angiogram of an 80-year-old patient during the second wave (December 2020) with multiple lung thromboembolism mainly located in the segmentary pulmonary artery branches; in image (**a**) some defect of opacification compatible with thrombi in some subsegmental branches (yellow arrow) of left superior lobes and in the segmentary branch for the apicodorsal segment (yellow arrow) in image (**b**); in image (**c**) lung thromboembolism in the lobar branch for the middle lobe (yellow arrow); and in image (**d**) in a subsegmental branch in the left inferior lobe (yellow arrow).

**Figure 24 biomedicines-11-02113-f024:**
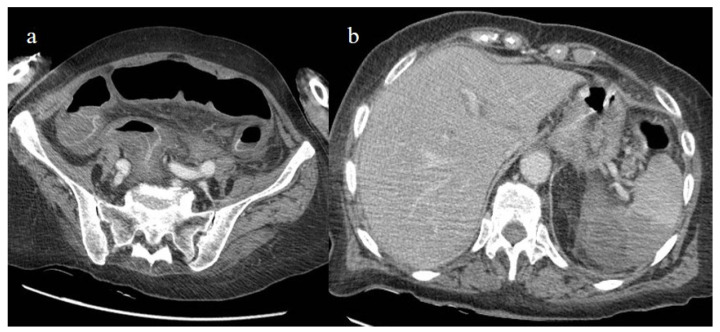
The enhanced computed tomography of an 80-year-old patient during the second pandemic wave (December 2020) with the findings in image (**a**) of diffuse colic thickness with target appearance and some hemorrhagic changes compatible with ischemic colitis; in image (**b**), a large ischemic area of the spleen is also present.

**Figure 25 biomedicines-11-02113-f025:**
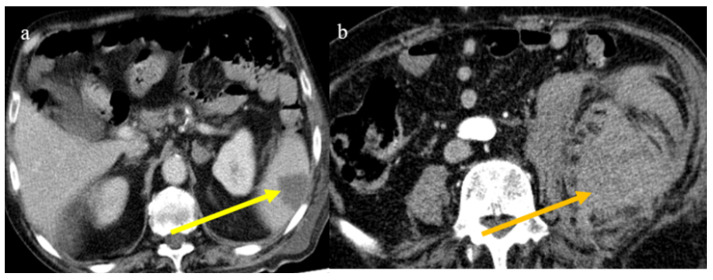
Enhanced computed tomography of an 80-year-old patient during the second pandemic wave showing an ischemic lesion of the spleen in image (**a**) (yellow arrow); the patient also had a large hematoma in the left psoas muscle, depicted in image (**b**) (orange arrow).

**Figure 26 biomedicines-11-02113-f026:**
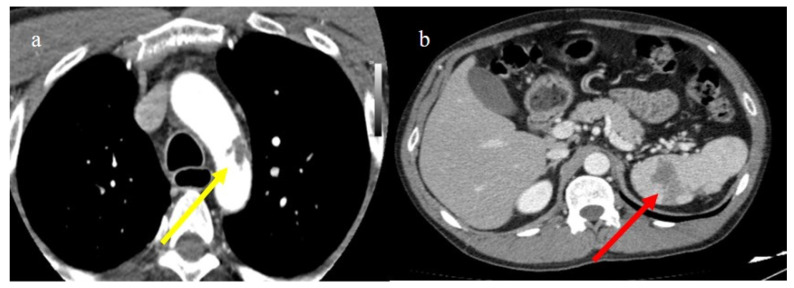
Enhanced computed tomography of a 61-year-old patient with a floating thrombus in the thoracic aorta at the level of the arch image (**a**) (yellow arrow) and with multiple vascular thromboembolism and ischemic areas in the spleen as depicted in the image (**b**) (red arrow). The patient had no known thromboembolic risk and had the second dose of m-RNA vaccine a few days earlier.

**Figure 27 biomedicines-11-02113-f027:**
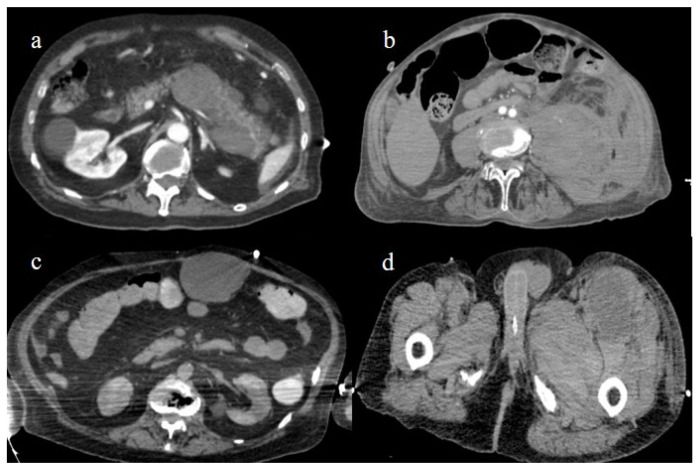
Enhanced computed tomography during the first and second pandemic waves, showing in image (**a**) a retroperitoneal hemorrhage involving the pancreas; in image (**b**) a large hematoma of the left psoas muscle; in image (**c**) hematoma of the rectus abdomins; and in image (**d**) a depiction of a hematoma in the muscles in the left thigh.

**Table 1 biomedicines-11-02113-t001:** This table summarizes the typical, indeterminate, and atypical appearances of COVID-19 pneumonia; ground glass opacities (GGOs).

**Typical Appearance**	GGOs with a crazy-paving pattern and consolidations in a peripheral and posterior or central-peripheral distribution; multilobar involvement; vascular enlargement, the halo and reversed halo sign; subpleural and parenchymal bands; and architectural distortion. They were predominant since the Delta wave.
**Indeterminate Appearance**	GGOs and consolidations with a unilateral, central, or upper-lobe distribution.
**Atypical Appearance**	Lobar consolidation, lung nodules or masses, miliary patterns, tree-in-bud patterns, cavitation, pleural effusion, central distribution, and lymphadenopathy. Atypical appearances were predominant during the Omicron waves.

**Table 2 biomedicines-11-02113-t002:** This table summarizes the main manuscripts that compared some chest CT features during different waves of the pandemic.

Chest CT Findings According to the Virus Variant	Virus Variant	Authors	Chest CT Features
Typical appearance	From the wild-type to the Delta variant.	Askani et al., 2022 [67]	The Delta variant presented more frequent typical features with more extensive lung involvement than the Omicron variant. The Omicron variant was more frequently associated with the absence of pneumonia.
		Inui et al., 2021 [68]	Typical findings were characteristic of the wild type to the Delta variant. GGOs with consolidation and repair changes were more frequent in the Delta variant. The Delta variant also showed more rapid pneumonia progression than the wild-type and Alpha variants.
		Ito et al., 2022 [54]	Peripheral GGO distributions were more frequent in the Alpha and Delta variants than the Omicron variant.
		Kirka et al., 2022 [27]	Typical features were found in 40.8% of patients with the wild-type variant and 1.7% of patients with the Omicron variant.
		Lee et al., 2023 [28]	Typical CT patterns were more frequent in the Delta group (76%) than in those with the Omicron variant (42%).
		Yang et al., 2022 [78]	Of patients with the Alpha variant, 86.84% presented typical COVID-19 pneumonia CT features.
		Yoon et al., 2023 [69]	Only 32% of patients with the Omicron variant presented typical findings, compared with 57% of the Delta variant cases.
Indeterminate appearance	Omicron variant	Ito et al., 2022 [54]	Cluster-like GGOs in the Omicron wave.
Atypical appearance	Omicron Variant	Hang et al., 2023 [75]	Patients infected with the Omicron variant presented a significantly higher prevalence of nodules, tree-in-bud patterns, and halo signs than patients with the original strain.
		Ito et al., 2022 [54]	Prevalence of non-peripheral distribution with random distribution during the Omicron wave.
		Lee et al., 2023 [28]	Peribroncovascular pneumonia with the Omicron variant and lower rates of severe pneumonia than the Delta variant.
		Tsakok et al., 2023 [77]	Patients with an Omicron infection presented a greater frequency of bronchial wall thickening but less severe disease compared with the Delta variant.
		Yang et al., 2022 [78]	Only 1.3% of patients infected with the Omicron variant had foci of pneumonia, and the GGOs were unilateral and centrilobular.
		Yoon et al., 2023 [69]	Peribroncovascular GGOs or centrolobular foci during the Omicron wave with less extensive pneumonia.

**Table 3 biomedicines-11-02113-t003:** This table summarizes the main lung complications and extrapulmonary vascular abdominal complications during the COVID-19 pandemic (ARDS: Acute respiratory distress syndrome; ICU: intensity care unit; FU: follow-up; PMS: pneumomediastinum; PTX: pneumothorax).

Lung Complications		Extrapulmonary Vascular Abdominal Complications	
COVID-19 Pneumonia	Secondary Lung Complications	Ischemic	Hemorrhagic
Typical findings of interstitial pneumonia with peripheral or peripheral central distribution from the wild/type variant since Delta variant	ARDS: Frequent occurrence since the Delta variant; some rare case reported as ADE phenomenon PMS and PTX: more commonly found as a consequence of barotrauma or non-invasive mechanical ventilation and more frequent during the first two pandemic waves Lung thromboembolism: more commonly involved the segmentary artery branches and was more common in the first two pandemic waves in patients in ICU Fibrotic changes: real fibrotic changes were rarely found ad 12 month of FU; fibrotic-like changes were more frequently found in patients with severe disease	Intestinal ischemia, ischemic colitis; parenchymal organ ischemia (Spleen and renal infarcts); all of these complications were rare and reported mainly during the first two waves of the pandemic	Spontaneous hematoma were more frequently located in the retroperitoneum and muscular abdominal wall; rare complications and more frequently reported during the first two pandemic waves as a consequence of anticoagulant treatment and consumption of coagulations factors
Absence of pneumonia or atypical findings during the Omicron wave			
Possible severe forms of COVID-19 pneumonia in breakthrough infections in the elderly and in patients with an immunosuppressive state			

## Data Availability

Data sharing not applicable.

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
