# Peer review of "A Pictorial Essay Describing the CT Imaging Features of COVID-19 Cases throughout the Pandemic with a Special Focus on Lung Manifestations and Extrapulmonary Vascular Abdominal Complications"

_biomedicines, 2023, doi:10.3390/biomedicines11082113_

Round 1
Reviewer 1 Report
The paper is really well written and is easily accessible. Despite do not contain original. Although it does not contain original ideas, the article offers a broad and satisfactory overview of the secondary events of SARS-CovII virus infection based on CT radiological data. The comparison between the variants of the virus is also interesting. In conclusion, the paper offers a simple but comprehensive review of the literature and can be considered for publication.
Some paragraphs contains different characters size. Figures need to be better inserted in the text and aligned with relatives descriptions.
English is fluent with very clear sentences. Only minor revision may be helpful
Author Response
Dear Reviewer,
Thank you for your valuable comments. The previous manuscript has been already edited however excluding the Figure legends by the MDPI service for the English language. However, the revised manuscript has been edited again for the English language by an English Professor. All the modifications are made in red. The Figures legends have been edited in English. In order to make the review more interesting, also following the suggestions of the second reviewer, we have also included a section on the pathogenesis focalizing on the role of the gut and lung axis for the pulmonary and extrapulmonary manifestations. We believe in the important role of this axis and also of its relationship with the innate and the adaptive immune systems. The section of the pathogenesis was indicated with the number 2 and was reported in red. We have also added in the section of lung complications, section 3.1 the differential diagnosis of COVID-19 pneumonia with the other viral pneumonia. We have also ordered the number of references and created the Table 3 that summarized this review. For our knowledge there aren’t a lot of reviews who summarize all features of COVID-19 pandemic, including the CT-features of breakthrough infections. On the other hand, we have also including the extrapulmonary vascular complications with a lot of images, that usually were but possible complications during the COVID-19 pandemic. We have also aligned the images, following your suggestions.
We hope that you will find all the modifications satisfactory.

Reviewer 2 Report
The author summarize the computed tomography (CT) imaging features of COVID-19 pneumonia and vascular extrapulmonary complications, including both ischemic and hemorrhagic abdominal complications.
The main limitation of this work is the lack of novelty. The authors simply describe the CT manifestations of Sars-Cov2.
I would suggest comparing the manifestations of Sars-Cov2 with other similar viruses to help a possible differential diagnosis.
It would also be useful to include the mechanisms underlying the manifestations of the virus.
The article is well structured but I would suggest a review summary table to aid reading.
The images are fine.
Author Response
The author summarize the computed tomography (CT) imaging features of COVID-19 pneumonia and vascular extrapulmonary complications, including both ischemic and hemorrhagic abdominal complications.
The main limitation of this work is the lack of novelty. The authors simply describe the CT manifestations of Sars-Cov2.
I would suggest comparing the manifestations of Sars-Cov2 with other similar viruses to help a possible differential diagnosis.
It would also be useful to include the mechanisms underlying the manifestations of the virus.
The article is well structured but I would suggest a review summary table to aid reading.
The images are fine.
Dear Reviewer,
Thank you for your valuable comments. In order to make the review more interesting and original, also following your comments about the poor of novelty, we have also included a section on the pathogenesis focalizing on the role of the gut and lung axis for the pulmonary and extrapulmonary manifestations in COVID-19. We believe in the important role of this axis and also of its relationship with the innate and the adaptive immune systems also for the development of new therapeutic strategies.
The section of the pathogenesis was indicated with the number 2 and was reported in red. We have also added in the section of lung complications, section 3.1 the differential diagnosis of COVID-19 pneumonia with the other viral pneumonia.
We have also added on the base of your suggestions the Table 3 that is a summary table. The Table 3 is reported at the end of the manuscript.
We have also ordered the number of references. For our knowledge there aren’t a lot of reviews who summarize all features of COVID-19 pandemic, including the CT-features of breakthrough infections. We have also included a lots of CT-images.
On the other hand, we have also included the extrapulmonary vascular complications reporting also a lot of images, that usually were but possible complications during the COVID-19 pandemic.
All the modifications are marked in red.
The manuscript has been Edited for the English language by an English Teacher. The original version has been previously edited by the MDPI English language service excluding the figures legends that now they were also edited.
We hope that you will find all the modifications satisfactory.
Round 2
Reviewer 2 Report
The manuscript has been sufficiently improved to warrant publication in Biomedicines.